# Marine Lectins and Lectin-like Proteins as Promising Molecules Targeting Aberrant Glycosylation Signatures in Human Brain Tumors

**DOI:** 10.3390/md22120527

**Published:** 2024-11-22

**Authors:** Ivan Buriak, Vadim Kumeiko

**Affiliations:** 1School of Medicine and Life Sciences, Far Eastern Federal University, 690922 Vladivostok, Russia; cutekasatik@gmail.com; 2A.V. Zhirmunsky National Scientific Center of Marine Biology, Far Eastern Branch, Russian Academy of Sciences, 690041 Vladivostok, Russia

**Keywords:** bivalve, cancer, glioma, diagnostics, lectin, C1qDC, *N*-glycan, *O*-glycan

## Abstract

Glycosylation is a ubiquitous and the most structurally diverse post-translational modification of proteins. High levels of phenotypic heterogeneity in brain tumors affect the biosynthetic pathway of glycosylation machinery, resulting in aberrant glycosylation patterns. Traditionally, unique glycocode readers, carbohydrate-binding proteins, have been used to identify differentially expressed carbohydrate determinants associated with the tumor cell surface. However, identifying novel distinctive glycosylation signatures in brain tumors requires the timely development of molecular tools capable of targeting them. We classified marine-derived lectins and lectin-like molecules according to their ability to cover aberrant glycosylation patterns in brain tumors to encourage exploration of the potential of these molecules for precision diagnostics and personalized therapy.

## 1. Introduction

One of the most common and structurally diverse post-translational modifications of biomolecules is glycosylation. Enzymatic addition of sugar residues to biopolymers has been found in all domains of life. Currently, researchers suggest that it is the recognition of molecular carbohydrate patterns on the cell surface that underlies the mechanisms of stimulus perception and response in physiological and pathological conditions [1].

It is known that cells in different physiological and pathophysiological conditions express differential glycan motifs. Malignantly transformed cells are characterized by significant variations in the glycosylation profile of cell membranes, forming localized areas of high-density carbohydrate clusters on the surface of tumor cells recognized by carbohydrate-binding proteins and thus mediating the modulation of cellular signaling, leading to tumor cell growth, migration, invasion, and chemoresistance [2]. Because lectins have a unique ability to selectively and highly specifically recognize a variety of glycoconjugates expressed by tumor cells, modern molecular tools for immunotherapy are currently being developed on their basis. Thus, lectins are used in CAR-T constructs [3,4] and lectibodies [5,6,7] for targeting and destruction of tumor cells.

Brain tumors are the most lethal and difficult to treat [8]. The heterogeneous etiology of brain tumors leads to diverse phenotypic manifestations, which are caused by epithelial–mesenchymal transition [9], genomic rearrangements, and, as a consequence, chromosomal instability [10], induction of a hypoxic state [11] and dynamic remodeling of the extracellular matrix [12], which contribute to rapid tumor progression and the emergence of drug resistance. Recently, evidence has emerged that dysregulation of the glycosynthetic apparatus in brain tumors leads to differential expression of glycosyltransferases involved in the biosynthesis of glycan structures on target proteins [13], which leads to the emergence of characteristic carbohydrate patterns dictated by the excessive function of glycosyltransferases. More detailed studies have already linked the altered expression of glycosyltransferases with the tumorigenic properties of glioblastoma [14,15]. This suggests that brain tumors can be differentially labeled according to their glycome profile features. The most malignant and untreatable brain tumors are gliomas, so most studies are aimed at identifying aberrant glycosylation in these neoplasms.

## 2. Brief Survey of the Classification of Marine Lectins and Lectin-like Molecules

For a long time, it was believed that carbohydrate-binding proteins are only lectins, but with the development of molecular biology and bioinformatics methods, as well as extensive accumulation of data on amino acid sequences and three-dimensional folds of carbohydrate-binding proteins, it became obvious that, in addition to the conservative carbohydrate recognition domain (CRD) of lectins, there are other evolutionarily non-homologous domains that perform the function of recognizing glycoconjugates. A huge repertoire of carbohydrate-binding proteins in marine organisms is the main system of innate immunity required for the recognition of pathogen-associated molecular patterns (PAMPs), which are a variety of carbohydrate components of the pathogen cell surface [16]. However, the greatest diversity of carbohydrate-binding proteins is found in the phylum Mollusk and the class Bivalves in particular. Currently, the main pattern recognition receptors (PRRs) in mollusks are the following families of carbohydrate-binding proteins: lectins, fibrinogen-related proteins (FREPs), C1q domain-containing proteins (C1qDCs), Toll-like receptors (TLRs), and scavenger receptors (SRs) [17].

Lectins are the best-characterized family of carbohydrate-binding proteins. According to the structure of highly conserved CRD regions in amino acid sequences and the structural similarity of protein scaffolds, lectins of the C-type, R-type, F-type, H-type, I-type, P-type, L-type, M-type, galectins, selectins, pentraxins, and ficolins are distinguished [18,19]. Fibrinogen-related proteins (FREPs) contain one or more homologous domains to fibrinogen (FBG) and have high amino acid sequence identity with C-terminal regions of the β- and γ-chains of fibrinogen. These proteins are ubiquitous in invertebrates and vertebrates. FREPs are involved in the immune response via their carbohydrate-binding properties [20]. Human C1q protein is a component of the classical pathway of the complement activation system, and it is the main link between innate and adaptive immunity [21]. C1qDCs contain the homologous domain of human C1q, and they have high structural homology to the B chain of human C1q protein. C1qDC proteins are unexpectedly widespread molecules among many invertebrates, despite their lack of a classic complement system. C1qDC proteins, members of the C1q/TNF superfamily, are widely distributed in invertebrates. Despite limited data on their structure and properties, their ability to interact with mammalian immunoglobulins and recognize aberrant glycosylation on human cells suggests significant potential for these molecules to develop new biotechnological and biomedical technologies [22].

TLRs are predominantly localized on the cell surface or in intracellular compartments (e.g., the endoplasmic reticulum membrane). Each Toll-like protein consists of a leucine-rich repeat (LRR) domain that mediates recognition of PAMPs, a transmembrane domain, and a cytoplasmic TIR tail that initiates downstream signaling [23]. SRs bind to components of the cell walls of Gram-positive and Gram-negative bacteria, as well as fungi (e.g., lipoteichoic acid, lipopolysaccharide, β-glucan), and induce the development of an immune response. The specificity of SRs to carbohydrate components of the pathogen cell surface is due to the presence of a conserved cysteine-rich SRCR domain at C-terminus of the protein [24].

## 3. Distinctive Features of *N*- and *O*-Glycosylation in Brain Tumors

Glycosylation is a complex and multistep process that requires the coordinated work of hundreds of glycosyltransferase enzymes, glycosidases, and glycosylation-related genes, and it affects several cellular compartments. The main types of glycosylation are *N*- and *O*-glycosylation, and they are also the most studied and characterized, so we pay close attention to them in our review. *N*-glycosylation begins in the endoplasmic reticulum with the synthesis of oligomannose structures that are attached to asparagine residues on target proteins via the core chitobiose linker. Proteins with oligomannose glycans are then transported to the Golgi apparatus, where the α1,2-mannosidases are used to trim the α1,2-linked mannose residues, resulting in the formation of mannopentose glycans (Figure 1a). At this stage of the glycosylation process, a marker carbohydrate signature already appears in gliomas due to the altered expression of α1,2-mannosidase MAN1A1. Reduced expression of MAN1A1 in glioma stem cells leads to the synthesis of high-mannose *N*-glycan on the CD133 glycoprotein, which is required for the CD133-DNMT1 interaction, which mediates the maintenance of glioma cells in the stemness state, enhancing drug resistance and tumorigenesis [25]. In addition, oligomannose structures with a predominance of the oligomannose glycan M5A were detected in glioblastoma tissue and cell lines propagated from them. Less mature oligomannose structures, M6B and M9A, were also detected [26]. Relatively recently, researchers have drawn attention to a special type of *N*-glycosylation called paucimannosylation, which is a truncated *N*-glycan structure [27]. Thus, a high level of paucimannosylation in U-87 MG and U-138 MG glioblastoma cell lines with a predominant glycan structure Man_3_GlcNAc_2_Fuc, which consists of a naked trimannosyl core and a fucosylated N-acetylglucosamine residue in the chitobiose core and is processed by the enzymes α-mannosidases 2 from the MAN2 family and α1,6- fucosyltransferase FUT8, is associated with proliferation, migration, and invasion of tumor cells [28].

After trimming of mannose residues to M5 oligomannose glycan, the formation of hybrid and complex glycans begins, the first step of which is carried out by enzymatic addition of a β1,2-linked N-acetylglucosamine to a mannose residue on the α1,3Man arm of the glycan by the enzyme N-acetylglucosaminyltransferase 1, which is encoded by the MGAT1 gene (Figure 1b). The β1,2-GlcNAc branch in hybrid glycans is subject to further elongation of the glycan structure. MGAT1 is highly expressed in glioblastoma samples and promotes complex *N*-glycosylation of Glut1, leading to proliferation and migration of glioma cells [29]. Hybrid *N*-glycans can be processed into biantennary *N*-glycans, which are a template for the formation of complex *N*-glycans. This process begins with the removal of the terminal α1,3- and α1,6-linked mannose residues on the α1,6Man arm and the subsequent addition of β1,2-GlcNAc residue to this arm. The formation of biantennary *N*-glycans is processed by the enzymes MAN2 family and *N*-acetylglucosaminyltransferase II (MGAT2). Interestingly, biantennary galactosylated *N*-glycan with core fucosylation (A2G2F glycan), which are synthesized by adding terminal residues of β1,4-linked galactose to the β1,2-GlcNAc branches and α1,6-linked fucose to the first GlcNAc residue in the chitobiose core using the enzymes α1,6-fucosyltransferase (FUT8) and N-acetylgalactosaminyltransferases from the B4GALT family, turned out to be elevated in glioblastoma tissue samples and glioma cell lines compared with normal brains. Intriguingly, A2G2F is expressed in the brain only during embryonic development [26]. Notably, the carbohydrate epitope A2G2F plays an important role in the metastasis of B16 melanoma cancer cells and the formation of pulmonary metastatic nodules, suggesting that the expression of the A2G2F glycan by glioma cells may influence their metastatic potential [30].

Then, sequential addition of β1,4- and β1,6-GlcNAc to α1,3- and α1,6-Man arm glycan occurs, respectively, forming tri- and tetra-antennary complex *N*-glycans (Figure 1c). Branching of the biantennary glycan occurs with the help of the enzymes N-acetylglucosaminyltransferase IV (MGAT4) and N-acetylglucosaminyltransferase V (MGAT5). Previously, researchers linked the presence of β1,6-GlcNAc branches on *N*-glycans to tumor metastasis [31]. β1,6-GlcNAc-bearing complex *N*-glycans involved in glioblastoma migration and invasiveness were studied by Yamamoto et al. Stable transfection of MGAT5 into human glioma cells resulted in a marked increase in the invasiveness of glioma in vitro [32]. It has been shown that increased expression of MGAT5, which mediates *N*-glycosylation of receptor protein tyrosine phosphatase type μ (RPTPμ) with β1,6-GlcNAc-branched *N*-glycans, reduces its activity, thereby increasing the invasiveness of glioma cells [33]. Interestingly, altered *N*-glycosylation patterns affect the glycocalyx thickness of cells, through which mechanosensing may be regulated. Thus, the expression of MGAT5, which catalyzes the formation of β1,6-GlcNAc-branched *N*-glycans, acts as a key player in ensuring migration in glioblastoma stem cells [34].

After the formation of branched *N*-glycans, the extension of the terminal GlcNAc-bearing branches begins, which is usually initiated by the addition of β1,3-4-linked galactose residues. Expression of β1,4-galactosyltransferase V (B4GalT-5), which catalyzes the transfer of β1,4-linked galactose to the β1,6-GlcNAc arm of *N*-glycans, forming β1,4-Gal-containing *N*-glycans, suppresses etoposide-induced apoptosis in glioma cells, testifying to its involvement in the development of drug resistance [35]. Also, overexpression of B4GalT-5 significantly suppressed arsenic trioxide (As_2_O_3_)-induced apoptosis in glioma cells, suggesting the use of B4GalT-5 inhibitors in combination therapy for gliomas [36]. In addition, it was noted that decreased B4GalT-5 expression in glioma cells caused a decrease in polylactosamine synthesis and selectively depleted CD133^+^ cells in the xenograft, thereby suppressing the ability of glioma cells to engage in self-renewal, which suggests an interesting role for B4GalT-5 in the formation of polylactosamine on *N*-glycans, which appears to be one of the molecular carbohydrate signatures of glioma stem cells [37].

Further elongation of *N*-glycans is associated with the addition of terminal capping residues of N-acetyllactosamine, sialic acid, and fucose with the formation of (poly)LacNAc-, (poly)Sia-, and fucosylated complex *N*-glycans. It is known that β1,3-N-acetylglucosaminyltransferase-8 (β3GnT8) is involved in the synthesis of the polylactosamine chain on β1,6-GlcNAc branched *N*-glycans (Figure 1d). β3GnT8 has increased expression in glioma tissues; in addition, an increase in β3GnT8 directly correlated with the grade of glioma malignancy. Suppression of β3GnT8 contributed to a decrease in the level of polylactosamine formation and demonstrated a decrease in the growth, migration, and metastatic potential of glioblastoma cells in vitro and in vivo [38]. Many tumor cells are characterized by hypersialylation and synthesize structurally diverse *N*- and *O*-glycan structures with different sialic acid content on their surface, including in brain tumors (Figure 1e) [39]. Thus, transfection of glioma cells U-373 MG α2,3-sialyltransferase III (ST3GAL3), which increases the content of terminal α2,3-linked sialic acids, resulted in a phenotype of glioma cells with increased invasiveness. Notably, transfection of the same cells with α2,6-sialyltransferase I (ST6GAL1), which catalyzes the addition of α2,6-linked sialic acids, suppressed glioma cell invasion. In addition, glioma cells expressing ST6GAL1 did not initiate tumors in vivo, suggesting that the abundance of terminal α2,6-linked sialic acids is a negative regulator of glioma cell invasion [40]. However, more recently, researchers have found that glioblastoma stem cells have higher levels of ST6GAL1 expression, which was associated with increased proliferation and self-renewal, as confirmed by ST6GAL1 knockdown, which resulted in decreased tumorigenic potential in glioblastoma stem cells via decreased levels of α2,6 sialylation of PDGFRB [41]. These data suggest that glycosylation by α2,3-, α2,6-, and α2,8-linked sialic acids is a dynamic process dependent on the molecular microenvironment of tumor cells, and it may lead to different phenotypic manifestations depending on the state of the cell during differentiation. High-grade astrocytomas highly express polysialic acid (PSA), while low-grade astrocytomas maintain low levels of PSA [42]. Because PSA is one of the post-translational modifications for the neural cell adhesion molecule (NCAM), an attempt was made to elucidate the effect of this type of glycosylation on the migratory potential of glioma cells. Thus, overexpression of PSA in rat glioma cells C6 showed an increased ability to invade into the corpus callosum. This suggests that NCAM decorated with PSA has an impaired ability to engage in cis- and trans-NCAM-NCAM interactions, thus facilitating glioma cell invasion [43]. Low-grade glioma cell lines showed low expression of all Lewis glycans (SLe^X^, Le^X^, SLe^a^, Le^a^, Le^y^, and Le) and truncated O-GalNAc glycans (Tn, STn, and T). SLe^X^ expression showed an association between high-grade glioma cell lines (only for U-87 MG (medium) and for LN-18 (low)). Expression of truncated O-GalNAc glycans was low for all glioma cell lines, but for U-87 MG and T98G, Tn expression was moderate. SW 1088 and Hs 683 showed no significant difference in binding to ConA and PHA-L, while LN-229, LN-18, U-373 MG, T98G, U-118 MG, and A-172 bound more to PHA-L, which recognizes the multi-antennary *N*-glycans presenting β1,6-GlcNAc complex. T98G and LN-229 showed high SLe^X^ expression and high binding to PHA-L, as well as high MGAT5 expression. Also, T98G and LN-229 expressed glycosyltransferases necessary for Lewis antigens synthesis. MGAT5 silencing reduced terminal SLe^X^ expression in *N*-glycans, thus changing adhesion and migration properties in high-grade glioma cells (Figure 1f) [44]. Complex *N*-glycans are also characterized by α1,6-fucosylation of the first GlcNAc residue in the chitobiose core. FUT8 is the only enzyme in the fucosyltransferase family that catalyzes the addition of α1,6-linked fucose to core *N*-glycans [45]. FUT8 is highly expressed in patient-derived tumor tissues and glioblastoma cells. Overexpression of FUT8 promotes aberrant fucosylation of receptor tyrosine kinase, which results in a malignant cell phenotype leading to extensive cell growth, migration and invasion, and stimulates the development of resistance to temozolomide [46].

*O*-glycosylation occurs in the Golgi apparatus and begins with the addition of N-acetylgalactosamine to serine or threonine residues in the target protein (Figure 1g). The researchers found that polypeptide N-Acetylgalactosaminyltransferase 12 (GALNT12) expression was highly linked with poor prognosis in glioblastoma, demonstrating that GALNT12 in U-87 MG glioblastoma cells mediates cell proliferation, migration, and invasion through the PI3K/Akt/mTOR pathway [47]. However, this appears in gliomas after the initiation of *O*-glycosylation, so complete elongation of *O*-glycans does not occur, resulting in the formation of so-called truncated *O*-glycans. Such aberrant *O*-linked glycosylation patterns are implicated in gliomagenesis, as the α-GalNAc-terminal glycan Tn antigen, a truncated version of *O*-linked glycan structures, is expressed in patient-derived glioblastoma cell lines and tissues and is an attractor of immunosuppressive tumor-associated macrophages [48]. Also, researchers proposed to use stage-specific embryonic antigen 1 (SSEA-1/Le^X^) as a tumor-associated carbohydrate epitope for glioblastoma stem cells, as the population of SSEA-1^+^ (Le^X+^) cells exhibited high tumorigenic potential, which corresponds to the phenotypic characteristics of glioblastoma stem cells [49].

## 4. Promising Marine Lectins and Lectin-like Molecules to Target *N*- and *O*-Glycosylation in Brain Tumors

Marine organisms are attractive bioresources for the isolation of lectins and proteins with lectin-like activity because they have an impressive repertoire of carbohydrate-binding molecules encoded in their genomes. However, to date, only a few marine-derived carbohydrate-binding proteins have been tested for the treatment and targeting of brain tumor cells.

Thus, a lectin from the sea bass *Dicentrarchus labrax* called DIFBL belonging to the F-type lectins with carbohydrate specificity for fucose [50], when fused to a soluble Coxsackie virus receptor, promoted infection of U-87 MG glioblastoma cancer cells with adenovirus [51]. In addition, DIFBL caused proliferation suppression of hepatocellular carcinoma Hep3B and BEL-7404 cells, lung cancer A549 cells, and colorectal cancer SW480 cells. In hepatocellular carcinoma Hep3B cells, DIFBL induced apoptosis through the PRMT5-E2F-1 pathway, which controls cell migration and invasion [52,53]. We propose that DIFBL is able to distinguish between terminal and core fucosylated glycan structures on the surface of brain tumor cells.

Another aberrant glycosylation pattern identified is terminal sialylation, which can be targeted by a number of marine carbohydrate-binding proteins belonging to the C1qDC class. Carbohydrate-binding protein MkC1qDC isolated from the bivalve *Modiolus kurilensis* and characterized by carbohydrate specificity for sialic acid exerted an inhibitory effect on cervical cancer HeLa cells [54]. Sialic-acid-specific lectin-like protein HddSBL isolated from the marine snail *Haliotis discus discus*, which is a representative of C1qDC [22], in the composition of oncolytic virus caused toxicity in a glioblastoma mouse model based on rat glioma C6 cells [55]. Also, HddSBL delivered by adenovirus Ad.FLAG reduced viability in human non-small-cell lung carcinoma A549 and H1299 cells, hepatocellular carcinoma Hep3B cells, and colorectal adenocarcinoma SW480 cells [56]. Another representative C1qDC protein in marine invertebrates OXYL from marine star *Anneissia japonica* can bind to N-acetyllactosamine (LacNAc) type 2 (Galβ1,4GlcNAc) but does not recognize LacNAc type 1 (Galβ1,3GlcNAc), and it could recognize breast cancer BT-474, MCF-7, and T47D cells without exhibiting cytotoxicity [57]. Further study of the carbohydrate specificity of OXYL revealed that it recognizes complex *N*-glycans containing LacNAc type 2 with α2,3-linked sialic acid [58]. Because OXYL is able to recognize LacNAc type 2, it can also be positioned to target (poly)LacNAc-containing *N*-glycans, which appear to be a marker of glioma stem cells. C-type lectin AVL from marine sponge *Aphrocallistes vastus* inhibited by sialylated-mucin glycans in an oncolytic virus showed significant anticancer activity against colon cancer HCT116 cells, human glioblastoma U-87 MG cells, breast cancer 4T1-LUC cells, and hepatocellular carcinoma BEL-7404 cells [59]. Further studies of AVL lectin revealed that as part of an oncolytic virus, it induces metabolic reprogramming in hepatocellular carcinoma cells by stimulating the production of reactive oxygen species (ROS). It was shown that the development of ROS-induced oxidative stress stimulated viral replication and induced apoptosis in hepatocellular carcinoma PLC/PRF/5 and Huh7 cells [60].

To detect truncated *O*-glycan Tn antigen, which is a carbohydrate determinant of glioblastoma cells, it is advisable to use C-type lectin APL from starfish *Asterina pectinifera* with specificity for Tn antigen [61]. Also, as a therapeutic agent, APL was used to target hepatocellular carcinoma Hep3B, Huh7, and PLC/PRF/5 cells [62] and breast cancer MCF-7 and MDA-MB-231 cells [63].

Mannose-specific lectins can be used to detect oligomannose structures in *N*-glycans, which exhibit diverse structural patterns from M9 to M3-containing *N*-glycans. Thus, lectin KSL from red alga *Kappaphycus striatus*, recognizing high mannose *N*-glycans, showed antiproliferative effects against colorectal adenocarcinoma HT29 cells, cervical adenocarcinoma Hela cells, breast cancer MCF-7 cells, human lung adenocarcinoma SK-LU-1 cells, and human gastric adenocarcinoma AGS cells [64]. Lectin UPL1, isolated from the seaweed *Ulva pertusa*, belongs to a special class of mannose-specific proteins related to methanol dehydrogenase, as it shows specificity for N-acetyl-D-glucosamine, but it also exhibits specificity for high-mannose glycans [65]. The unique carbohydrate specificity of UPL1 lectin can be exploited to bind hybrid *N*-glycans. Adenovirus-delivered UPL1 lectin affected multiple signaling cascades in hepatocellular carcinoma Huh7 and BEL-7404 cells [66]. SfL lectin isoforms from the seaweed *Solieria filiformis*, possessing great similarity to the OAAH family of lectins, have specificity to branched mannopentose oligosaccharides and exerted an anticancer effect on breast cancer MCF-7 cells [67]. Lectin ESA from the seaweed Eucheuma serra with specificity for high mannose *N*-glycans exhibited anticancer effects on colorectal adenocarcinoma Colon26 cells in vitro and in vivo [68]. ESA also had a cytotoxic effect on cervical cancer HeLa cells and colorectal adenocarcinoma Colo201 cells. It turned out that breast cancer MCF-7 cells had a relatively high resistance to the cytotoxic effect of ESA. The mechanism of death of colorectal adenocarcinoma Colo201 cells included DNA fragmentation and apoptosis mediated by the interaction of ESA with high mannose glycans presented on the cell surface. ESA immobilized on lipid vesicles selectively recognized colorectal adenocarcinoma Colo201 cells but not human normal fibroblasts and non-tumorigenic immortalized breast epithelial MCF10-2A cells, which will allow for using ESA as an effective targeting agent for the recognition of mannose-rich glycans on the surface of tumor cells [69]. Mannose-specific lectin BPL2 isolated from the seaweed *Bryopsis plumosa* was shown to be effective against lung cancer A549, H460, and H1299 cells. Notably, BPL2 inhibits cancer cell proliferation by modulating epithelial–mesenchymal transition genes while demonstrating low cytotoxicity against normal lung MRC5 cells [70]. It is likely that D-mannose-specific BPL2 lectin will recognize the trimannosyl core and will be suitable for identifying M3-M1 paucimannose structures and, specifically, the Man_3_GlcNAc_2_Fuc epitope that has been found in glioblastoma cells.

Another marker of glioma stem cells, A2G2F glycan, should probably be detected using lectins specific to terminal galactose residues. N-acetyl-D-galactosamine/galactose (GalNAc/Gal)-specific lectin CGL from the bivalve *Crenomytilus grayanus* recognizes globotriose Gb3 (Galα1,4Galβ1,4Glc-Cer) on the surface of metastatic breast cancer MCF-7 cells and reduces their viability [71]. In addition, CGL exhibits cytotoxicity towards Burkitt’s lymphoma Raji cells, which are characterized by overexpression of Gb3, and it has no effect on erythroleukemia K562 cells, which do not have Gb3 on their surface [72]. Recent evidence suggests that CGL is a mytilectin, a unique subset of R-type lectins characterized by a β-trefoil fold [73]. Notably, another member of this lectin family, MytiLec, was also able to recognize Gb3 and eliminate Burkitt’s lymphoma Ramos cells, which are characterized by high levels of Gb3 expression [74]. Using synthetic bioengineering techniques, MytiLec was used to construct a Mitsuba-1 lectin consisting of three identical carbohydrate-recognizing subdomains that specifically labeled Burkitt’s lymphoma Raji cells expressing Gb3 without affecting cell viability [75]. Another GalNAc/Gal-specific lectin HCL from the marine sponge *Haliclona cratera* showed strong inhibitory activity on the growth of cervical cancer HeLa cells and metastatic melanoma FemX cells [76].

One of the problems with the use of marine carbohydrate-binding proteins for diagnostic purposes is their relatively low annotation of carbohydrate-binding activity toward specific glycan structures, i.e., *N*-glycans. We also hypothesize that groups of lectins and lectin-like molecules specific for N-acetyl-D-glucosamine (GlcNAc) will be useful in recognizing GlcNAc-terminated *N*-glycans or branched *N*-glycans. Lectin ACL-1 from the marine sponge *Axinella corrugata* can bind to GalNAc- and GlcNAc-bearing glycans. This lectin was used for differential staining of various carcinomas [77]. Lectin iNol-binding monosaccharides and glycoproteins with N-acetylated groups from sleeper lobster *Ibacus novemdentatus* exerted a cytotoxic effect on breast cancer MCF-7 and T47D cells, cervical cancer HeLa cells, and colorectal adenocarcinoma Caco-2 cells [78]. Lectin DTL from the ascidian *Didemnum ternatanum*, exhibiting specificity for GlcNAc, showed antiproliferative properties on cervical cancer HeLa cells [79].

Lectin HOL-18 from the marine sponge *Halichondria okadai* has carbohydrate specificity for complex *N*-glycans [80] and showed a cytotoxic effect on cervical cancer HeLa cells and breast cancer MCF-7 and T47D cells [81]. Complex *N*-glycans are an integral part of the glycosylation profile of glioma cells of varying degrees of differentiation, but additional studies are needed to determine the molecular phenotype that HOL-18 recognizes for its targeted use in brain tumor cell typing. General patterns of aberrant *N*- and *O*-glycosylation in brain tumor cells that can be recognized by marine-derived lectins and lectin-like molecules are summarized in Figure 2.

The above lectins and lectin-like molecules preferentially target tumor cells of carcinomas (cervical cancer, lung cancer, hepatocellular cancer, colorectal cancer, breast cancer, gastric cancer, melanoma), which develop from tissues of epithelial origin, while gliomas are aggressive malignancies arising from neuroepithelial tissue. Despite their scrambled molecular etiology and phenotypic heterogeneity, gliomas and carcinomas demonstrate a shared mechanism of carcinogenesis, undergoing an epithelial–mesenchymal transition (EMT), as a result of which epithelial cells acquire mesenchymal properties. EMT involves remodeling of the cellular cytoskeleton and disruption of cell–extracellular matrix adhesion and cell–cell interactions, which leads to the emergence of invasive and metastatic cellular behavior [9,82]. Recent data indicate a close correlation of EMT with disturbances in the glycosylation biosynthetic machinery, leading to the emergence of glycosylated neo-epitopes that dysregulate the recognition of carbohydrate patterns on the cell surface and cause abnormal modulation of signal transduction pathways [83].

A characteristic feature of carcinoma tumor cell glycosylation is the expression of α2,3-, α2,6-, and α2,8-terminal sialic acid residues within the glycans. Thus, α2,3-terminal sialic acid residues within *O*-glycans present on the CD44 receptor have been identified as a biomarker of breast cancer stem cells [84]. In addition, the expression of glycan structures bearing α2,6-terminal sialic acid residues has been identified as a hallmark of metastatic breast cancer cells [85]. An interesting feature of metastatic colorectal cancer cells is the prevalence of α2,6-linked sialic acid residues in *N*-glycans, while α2,6-linked sialic acid residues mediate integrin-dependent adhesion of hepatocarcinoma cells to fibronectin and also promote adhesion of melanoma cells to extracellular matrix components and the basement membrane [86,87,88]. In contrast, for melanoma, the presence of α2,3-linked terminal sialic acid residues was shown to be a critical regulator of metastatic melanoma cell dissemination [88]. Ganglioside GD2, carrying an α2,8-linked terminal sialic acid residue, initiates the process of metastasis formation via the FAK–AKT–mTOR signaling pathway [89]. Hypersialylated NCAM with α2,8-linked sialic acid residues initiated EMT in breast epithelial cells and promoted cell motility via the EGFR/STAT3 pathway [90]. Polysialylation is also characteristic of lung and colorectal cancer cells [91,92]. Aberrant mucin-type O-GalNAc glycosylation is characteristic of carcinomas. Despite this, a common feature of mucin glycosylation is a disturbance in the processes of elongation and terminal capping of core *O*-glycan structures, which contributes to the formation of truncated glycan structures, which are often associated with malignant transformation processes and correlate with the severity and prognosis of cancer [93]. The main truncated *O*-glycan of carcinomas is the Tn antigen (GalNAcα-O-Ser/Thr), which predetermines their malignant phenotype and metastatic potential [91,94,95,96]. Carcinomas are also characterized by an altered cell surface fucosylation status. Thus, reduced expression of α-L-fucosidase 1 (FUCA1) and increased expression of α1,6-fucosyltransferase (FUT8) lead to an increase in α1,3-linked fucose in the Lewis X antigen (Le^X^) and core residues of α1,6-linked fucose in *N*-glycans in breast cancer [97,98]. In addition, sialylated Lewis X antigen (SLe^X^) is a carbohydrate determinant of metastatic breast cancer [99]. The above-mentioned Lewis antigens are also characteristic of other carcinomas, thus playing an important role in their development and progression [86,87,91]. Also, such patterns of aberrant glycosylation, characterized by an abundance of α1,6-linked fucose, high-mannose branches, specific paucimannose glycans, A2G2F glycan and poly-N-acetyllactosamine chains ((poly)LacNAc) in the structure of β1,6-branched complex *N*-glycans confer a resemblance between carcinoma and glioma cells [30,86,87,100,101]. As a consequence, the glycosylation profiles of carcinomas and gliomas have overlapping molecular fingerprints representing carbohydrate determinants that underlie their malignant phenotypes. Intriguingly, common glycosylation patterns reflect the invasive and metastatic potential of carcinoma tumor cells, suggesting that these onco-associated glycans exposed on the glioma cell surface are involved in invasion and migration, thus enhancing the infiltrating phenotype. By identifying hotspots in the glycosylation landscapes of carcinomas and gliomas, we propose that lectins and lectin-like proteins specific to the tumor surface of carcinomas can be repositioned for selective recognition of gliomas, and after defining their fine carbohydrate specificity through the identification of preferred carbohydrate ligands, it is possible to create precision molecular labeling tools that differentiate glioma cell populations depending on the cell surface glycosylation status for improved diagnostics and the selection of further therapeutic strategies.

## 5. Conclusions

Lectins and lectin-like molecules of marine organisms have long been considered exclusively as molecules that provide innate immunity via opsonization facilitating phagocytosis. This paradigm limited their investigation in terms of recognition specificity that was predominantly focused on pathogen-associated molecular patterns and respective monosaccharides. Unfortunately, this approach has bypassed the study of the mechanisms of carbohydrate epitope recognition through which these molecules perform their regulatory and other immune functions. Researchers realized that lectins and lectin-like molecules are the main molecules capable of deciphering surface glycosylation patterns for cells of different origins, including microorganisms, animals, and humans. This piqued their interest in determining the fine carbohydrate specificity of these molecules and in attempting to differentiate the phenotypic manifestations of normal and tumor cells. These efforts led to the development of several highly specific lectin-based probes for enrichment or targeting of cell populations and subpopulations of different origins. In terms of cancer research, the tools for molecular phenotyping of carbohydrate determinants for highly heterogeneous tumors are of great interest, and brain tumors, due to their particular heterogeneity, with the highest carbohydrate content characteristic of the extracellular compartment, could be effectively recognized using a set of lectin probes. At present, glycomics of brain tumors is in its infancy, and significant efforts are required to identify aberrant glycosylation patterns in such tumor types. However, available literature data allow us to conclude that brain tumors have some distinctive glycosylation structures that are characteristic of the oncogenesis process. To detect them, it is advisable to use carbohydrate-binding molecules of marine organisms, which have a rich repertoire of such molecules. In our work, we summarize this research, and we call on the scientific community to conduct active research on carbohydrate determinants of brain tumors to finely characterize the carbohydrate specificity of lectins and lectin-like molecules of marine organisms to specifically select detecting probes for modern diagnostics and treatment of these malignant neoplasms. Here, we provide a prospective research plan fostering investigations into carbohydrate-binding proteins of marine organisms, some of which were tested on glioma cells without precise glycomics, and some others need to be probed due to their presumptive potential in recognizing diversified patterns of aberrant glycosylation in brain tumors.

## Figures and Tables

**Figure 1 marinedrugs-22-00527-f001:**
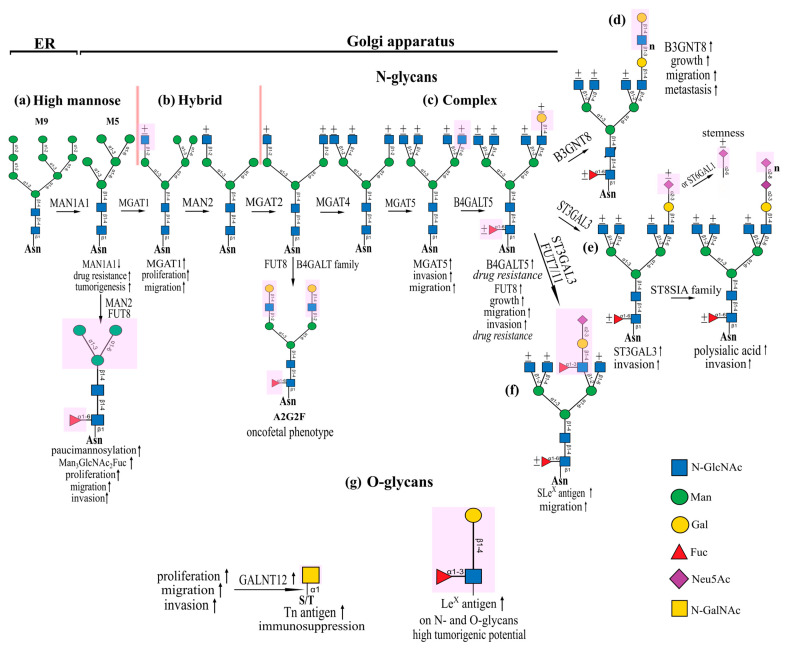
Aberrant glycosylation pathways in brain tumors. (**a**) High mannose *N*-glycans synthesis; (**b**) hybrid *N*-glycans synthesis; (**c**) complex *N*-glycans synthesis; (**d**–**f**) *N*-glycan terminal capping synthesis; (**g**) *O*-glycans synthesis.

**Figure 2 marinedrugs-22-00527-f002:**
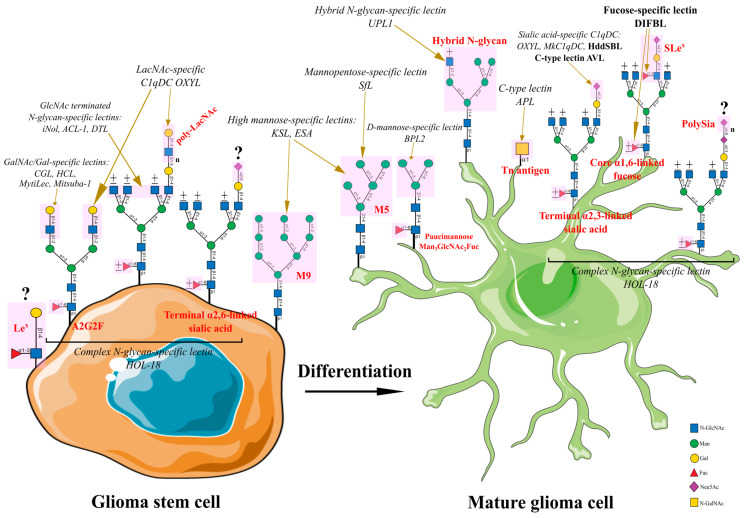
Carbohydrate antigens associated with brain tumors and proteins derived from marine organisms attributed to carbohydrate-binding activity promising for their detection.

## Data Availability

Data sharing is not applicable.

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
