# Peer review of "Marine Lectins and Lectin-like Proteins as Promising Molecules Targeting Aberrant Glycosylation Signatures in Human Brain Tumors"

_marinedrugs, 2024, doi:10.3390/md22120527_

Round 1
Reviewer 1 Report
Comments and Suggestions for Authors
The article provides a narrative review addressing the potentiality of marine lectins and lectin-like proteins for being employed in the diagnostics and personalized therapy of human brain tumors.
It is well written, including a good introduction showing the state of the art, as well as a comprehensive report on the distinctive features of N- and O-glycosilation with adequate illustrations. Nontheless, the discussion of the reported data should have been more extensive and the references more complete.
Based on the presumptive potential of marine lectins and lectin-like proteins in recognizing diversified patterns of the aberrant glycosilation present in brain tumors, the authors offer a classification of these molecules and propose encouraging research on this theme to apply them in the diagnosis and treatment of these tumors.
Author Response
Reviewer`s comment 1:
The article provides a narrative review addressing the potentiality of marine lectins and lectin-like proteins for being employed in the diagnostics and personalized therapy of human brain tumors.
It is well written, including a good introduction showing the state of the art, as well as a comprehensive report on the distinctive features of N- and O-glycosilation with adequate illustrations. Nontheless, the discussion of the reported data should have been more extensive and the references more complete.
Based on the presumptive potential of marine lectins and lectin-like proteins in recognizing diversified patterns of the aberrant glycosilation present in brain tumors, the authors offer a classification of these molecules and propose encouraging research on this theme to apply them in the diagnosis and treatment of these tumors.
Author’s comment:
Dear Reviewer 1, we would like to express our sincere gratitude for your feedback and constructive comments on our manuscript. Your insights have been invaluable in enhancing the quality of our work. In response to your suggestions, we have undertaken a comprehensive revision of the manuscript. Thus, in accordance with your comment we have revised Chapter 4 and added discussion elements to it. In this way, we found additional information regarding our proposed proteins for detecting aberrant glycosylation patterns of gliomas. It turned out that some of the mentioned carbohydrate-binding proteins have been further investigated in the context of their antitumor properties. In addition, your suggestion to improve the citation quality led to the identification of two more carbohydrate-binding proteins that can potentially be used to recognize glioma-associated glycans (MytiLec and Mitsuba-1) (see L 253–254, L 256–259, L 283–287, L 309–318, L 332–338 and Figure 2). In addition, we have introduced a detailed discussion element into our Сhapter 4 following your suggestion (see L 365–424). For this purpose, we «drew parallels» between the glycosylation patterns of carcinomas and gliomas. It turns out that carcinomas and gliomas share a number of common glycan patterns and structures. This relationship improves the prognostic value of our proposed carbohydrate-binding proteins. Furthermore, we have addressed minor issues, including spelling mistakes and mismatches, to ensure the accuracy and professionalism of the manuscript (see L 10, L 14-15, L 196, L 200-201, L 204-205, L 206-207, L 209, L 210, L 303-304). We appreciate your support and guidance.
Reviewer 2 Report
Comments and Suggestions for Authors
The proposed article is well written and correctly well developed, explaining in detail the glycomic aberrations that occur when the brain tumor glioma develops. Fig. 1 perfectly illustrates the description of the changes undergone at the level of O- and N-glycans, with the involvement of all the enzymatic machinery necessary for the production of new glycan structures with an increase in ficisilations and sialylations, as well as their involvement at the cellular level such as proliferation, migration and invasion, characterizing a poor prognosis. Point 3 describes the state of art of the in vitro studies for marine lectins, which exert antitumor activity due to exhibiting specificity for some of the characteristic glycomic motifs of these tumor lines. However, the authors begin this point by drawing a parallel between lectins and lectins-like proteins. Only after some reading does it become comprehensive, although it is not understood what the C1qDC class proteins are. For that: -It is recommend that the authors correctly establish the difference between lectins and lectin-like, explaining that the class of proteins mentioned C1qDC exist in mollusks, so that the difference can be perceived.
Author Response
Reviewer’s comment:
The proposed article is well written and correctly well developed, explaining in detail the glycomic aberrations that occur when the brain tumor glioma develops. Fig. 1 perfectly illustrates the description of the changes undergone at the level of O- and N-glycans, with the involvement of all the enzymatic machinery necessary for the production of new glycan structures with an increase in ficisilations and sialylations, as well as their involvement at the cellular level such as proliferation, migration and invasion, characterizing a poor prognosis. Point 3 describes the state of art of the in vitro studies for marine lectins, which exert antitumor activity due to exhibiting specificity for some of the characteristic glycomic motifs of these tumor lines. However, the authors begin this point by drawing a parallel between lectins and lectins-like proteins. Only after some reading does it become comprehensive, although it is not understood what the C1qDC class proteins are. For that: -It is recommend that the authors correctly establish the difference between lectins and lectin-like, explaining that the class of proteins mentioned C1qDC exist in mollusks, so that the difference can be perceived.
Author’s response:
Dear Reviewer 2, we would like to express our sincere gratitude for your feedback and constructive comments on our manuscript. Your insights have been invaluable in enhancing the quality of our work. In response to your suggestions, we have undertaken a comprehensive revision of the manuscript. Thus, in accordance with your comment, we have introduced a chapter (see L 53-95) devoted to the classification of carbohydrate-binding proteins from marine organisms, in which we introduce the definition of the class of C1qDC proteins possessing lectin-like activity. Furthermore, we have addressed minor issues, including spelling mistakes and mismatches, to ensure the accuracy and professionalism of the manuscript (see L 10, L 14-15, L 196, L 200-201, L 204-205, L 206-207, L 209, L 210, L 303-304). We appreciate your support and guidance.
Reviewer 3 Report
Comments and Suggestions for Authors
This a well-prepared review manuscript with a condensed but comprehensive summary of our current knowledge about glycosylation patterns in brain cancer and marine lectins available to detect these patterns.
Author Response
Reviewer’s comment:
This a well-prepared review manuscript with a condensed but comprehensive summary of our current knowledge about glycosylation patterns in brain cancer and marine lectins available to detect these patterns.
Author’s response:
Dear Reviewer 3, Thank you for your thoughtful and positive feedback on our manuscript. We greatly appreciate the time and effort you have devoted to reviewing our work. We are pleased to hear that you found our study valuable and that it will contribute to the development of new molecular tools based on carbohydrate-binding proteins for glioma diagnostics.